# Maximizing the clinical utility and performance of cytology samples for comprehensive genetic profiling

David Kim [1] ✉, Chad M. Vanderbilt [1], Soo-Ryum Yang[1],
Subhiksha Nandakumar[2,3], Khedoudja Nafa[1], Rusmir Feratovic[1],
Natasha Rekhtman[1], Ivelise Rijo[1], Jacklyn Casanova [1], Anita Yun[1],
A. Rose Brannon [1], Michael F. Berger [1,2,3], Marc Ladanyi [1,2], Oscar Lin[1] &
Maria E. Arcila [1]

Comprehensive molecular profiling by next-generation sequencing has revolutionized tumor classification and biomarker evaluation. However, routine implementation is challenged by the scant nature of diagnostic material obtained through minimally invasive procedures. Here, we describe our long-term experience in profiling cytology samples with an in-depth assessment of the performance, quality metrics, biomarker identification capabilities, and potential pitfalls. We highlight the impact of several optimization strategies to maximize performance with 4,871 prospectively sequenced clinical cytology samples tested by MSK-IMPACT™. Special emphasis is given to the use of residual supernatant cell-free DNA (ScfDNA) as a valuable source of tumor DNA. Overall, cytology samples are similar in performance to surgical samples in identifying clinically relevant genomic alterations, achieving success rates up to 93% with full optimization. While cell block (CB) samples have excellent performance overall, low-level cross-contamination is identified in a small proportion of cases (4.7%), a common pitfall intrinsic to the processing of paraffin blocks, suggesting that more stringent precautions and processing modifications should be considered in quality control initiatives. By contrast ScfDNA samples have negligible contamination. Finally, ScfDNA testing exclusively used as a rescue strategy, delivered successful results in 71% of cases where tumor tissue from CB was depleted.

Comprehensive tumor molecular profiling using next-generation sequencing (NGS) technology is steadily increasing in routine oncologic practice, in order to guide precise disease classification and the selection of targeted therapies[1,2]. Concurrently, minimally invasive procedures have also steadily and systematically become a dominant tumor sampling modality. Despite indisputable patient benefits, the amount of tissue procured through such procedures is limited, raising concerns about their sufficiency and suitability for comprehensive downstream analysis.

Cytologic specimens are among the most limited tissue samples that are obtained through minimally invasive procedures. While often the only source of tumor for both diagnostic and biomarker evaluation,

[1]Department of Pathology & Laboratory Medicine, Memorial Sloan Kettering Cancer Center, New York, NY, USA. [2]Human Oncology and Pathogenesis Program, Memorial Sloan Kettering Cancer Center, New York, NY, USA. [3]Kravis Center for Molecular Oncology, Memorial Sloan Kettering Cancer Center, New York, NY, USA. ✉e-mail: kimd8@mskcc.org

**Table 1 | Demographic information of study cohort**

| Characteristic | Overall, $N = 4871^1$ | CB, $N = 4588^1$ | ScfDNA, $N = 283^1$ | p value² |
|---|---|---|---|---|
| Age, median (IQR) | 66 (59, 74) | 66 (58, 74) | 67 (60, 74) | 0.4 |
| Sex | | | | 0.2 |
| Female, N (%) | 2639 (54%) | 2497 (54%) | 142 (50%) | |
| Male, N (%) | 2232 (46%) | 2091 (46%) | 141 (50%) | |
| Outside/internal | | | | <0.001 |
| Internal, N (%) | 3091 (63%) | 2809 (61%) | 282 (100%) | |
| Outside, N (%) | 1780 (37%) | 1779 (39%) | 1 (0.4%)* | |
| Procedure | | | | <0.001 |
| Fluid, N (%) | 693 (14%) | 633 (14%) | 60 (21%) | |
| FNA, N (%) | 4178 (86%) | 3955 (86%) | 223 (79%) | |

*IQR* interquartile range, *N* the number of samples. Categorical variables were compared with Pearson's Chi-squared test. For continuous variables, a Wilcoxon rank sum test was used. Two-sided statistical tests were applied. *Received in CytoLyt and processed Internally.

judicious protocols for tissue testing have been explored to maximize the material available for genetic studies. In this context, the performance of NGS across various preparations, including cellblocks (CB), smears, and liquid-based suspensions, have been studied and described in the literature primarily focusing on small gene panels (<100 genes)[3–11]. However, with the increasing need for the assessment of a wider range of genetic alterations, the sufficiency and high performance of comprehensive NGS assays have remained a major concern.

To address this, our pathology department embarked on a comprehensive performance improvement project which involved several years of sequential process optimization to strategically improve the use of cytologic tissue samples for molecular profiling. This encompassed coordinated changes by the cytology lab, including the use of a modified HistoGel-based cell-block processing to improve pellet density[12,13], as well as changes in the diagnostic molecular lab in tissue processing such as deparaffinization with mineral oil[14–16], improved bead-based extraction techniques, implementation of dual index sequencing and adjustments of minimum DNA input requirements (Supplementary Table 1). Throughout this time, we also implemented the use of residual cytology supernatant fluids as an additional source of tumor DNA for NGS applications. Commonly discarded in routine cytology practice, supernatant fluids contain variable amounts of DNA from fragmented cells as well as whole cells, denoted here-on as supernatant cell-free DNA (ScfDNA)[16]. The strategic use of this DNA enables the preservation of cellular tissue for other ancillary studies that rely on visual assessment of intact cells, such as immunohistochemistry and cytogenetics.

Here, we show our overall clinical experience across a wide range of tumor types using cytologic material for comprehensive NGS, integrating the use of ScfDNA as a rescue sample when another material is unavailable. We re-analyzed all cytology sample data collected from our institution-wide prospective sequencing effort using the Memorial Sloan Kettering-Integrated Mutation Profiling of Actionable Cancer Targets (MSK-IMPACT™) assay, an FDA-cleared, paired tumor-normal hybridization-capture based NGS test, designed to comprehensively assess mutations, copy number alterations, and select rearrangements[17]. A summary of the performance characteristics across years of process optimization is presented, describing the utility and potential pitfalls of cytology samples for the identification of clinically relevant biomarkers, with comparisons to existing sequencing data from biopsies and resections from the same patients.

## Results

### Clinicopathologic characteristics of cytology sequencing cohort

In total, 4871 cytology tumor samples from 4633 patients were received for MSK-IMPACT testing with patient demographics detailed

in Table 1. Most samples were from CB preparations, 94.2% (4588/4871) while 5.8% (283/4871) were received as ScfDNA. Of note, ScfDNA testing was requested only when no other material was suitable or available. The majority, 63%, were procured at MSKCC and processed internally, while 37% were submitted from outside institutions (Table 1). Testing was canceled on 3% (146/4871) prior to sample processing for logistical considerations, including the lack of a submitted normal control for matched testing or testing no longer relevant for patient management. A diverse array of tissue sites and sample types were profiled as detailed in Fig. 1a, b. The cohort encompassed 181 unique tumor types with lung and pancreatic adenocarcinomas being the most common. The number of samples and relative frequencies of the profiled tumor types are further summarized in Supplementary Data 1.

### Success rate of NGS testing on cytology samples

Overall, 81% (3806/4725) of all samples were successfully tested. The success rate was higher for CB (81%; 3616/4457) samples compared to ScfDNA (71%; 190/268), noting that ScfDNA samples encompassed only cases for which the CB had already been deemed unsuitable for any analysis. Across the study period, the use of ScfDNA as a rescue sample boosted the overall success rate of the cytologic procedures from 77% to 81%. Causes of failure, in descending order of frequency, included low DNA yield below the minimum cutoffs established for sequencing (11.3%), very scant tumor tissue (i.e. <10% tumor) seen on the manual review (4.6%), low sequencing coverage below a median of 50× (1.8%), high sample level DNA contamination (1.6%) and low DNA quality (0.1%) including adequate coverage but high background noise and low base quality scores. (Fig. 1c).

To assess the impact of optimization efforts implemented across the study period, all requests (excluding cancellations) were stratified by year and testing success status. Sequential and statistically significant improvements were observed, even in the context of the increased number of genes tested by panel updates, reaching 89% in the last year of assessment (Fig. 2a). For samples deemed sufficient for sequencing (following qualification for tumor content and DNA yield), success rates were consistently high across all years (range: 96-98%) (Fig. 2b).

Among CB preparations, success rates were significantly higher for internal samples compared to those from outside laboratories, with highest success rates of 92–93% (internal) and 79-82% (external) in the last 2 years, in accordance with the full optimization efforts ($p < 0.01$; Fig. 2c).

On average, successful cytology samples had higher tumor purities compared to samples that failed testing, across both preparations (CB: $p = 0.0003$, ScfDNA: $p = 0.63$; Fig. 2c). Median total DNA yields were 427.5 ng and 182.2 ng ($p = 2.2 \times 10^{-16}$) for CB and ScfDNA, respectively (Fig. 2d). The lower DNA yield in ScfDNA was expected as these constituted rescue samples when the corresponding cytology tissue was too scant or exhausted.

### Sequencing performance: total coverage and sample quality metrics

Among the 4725 samples sequenced, the total median coverage was 586x. Coverages were significantly higher for CB samples compared to rescue ScfDNA samples, at 595× and 263×, respectively ($p = 2.2 \times 10^{-16}$) (Fig. 3a). Despite the lower coverage, most rescue samples retained coverages above 200×, which is above our established requirements to maintain sensitivity for variants calling at 2%.

Notably, in 2021 the minimum DNA input requirement for MSK-IMPACT was lowered from 50 ng to 30 ng for cell blocks. Following this change, we saw no significant differences in sequencing coverage (Fig. 3b), and the sequencing success rate remained steady at 98% across sequenced cases.

Among all samples sequenced, contamination checks revealed clinically relevant non-patient DNA contamination (≥2%) in 5.2% of cases

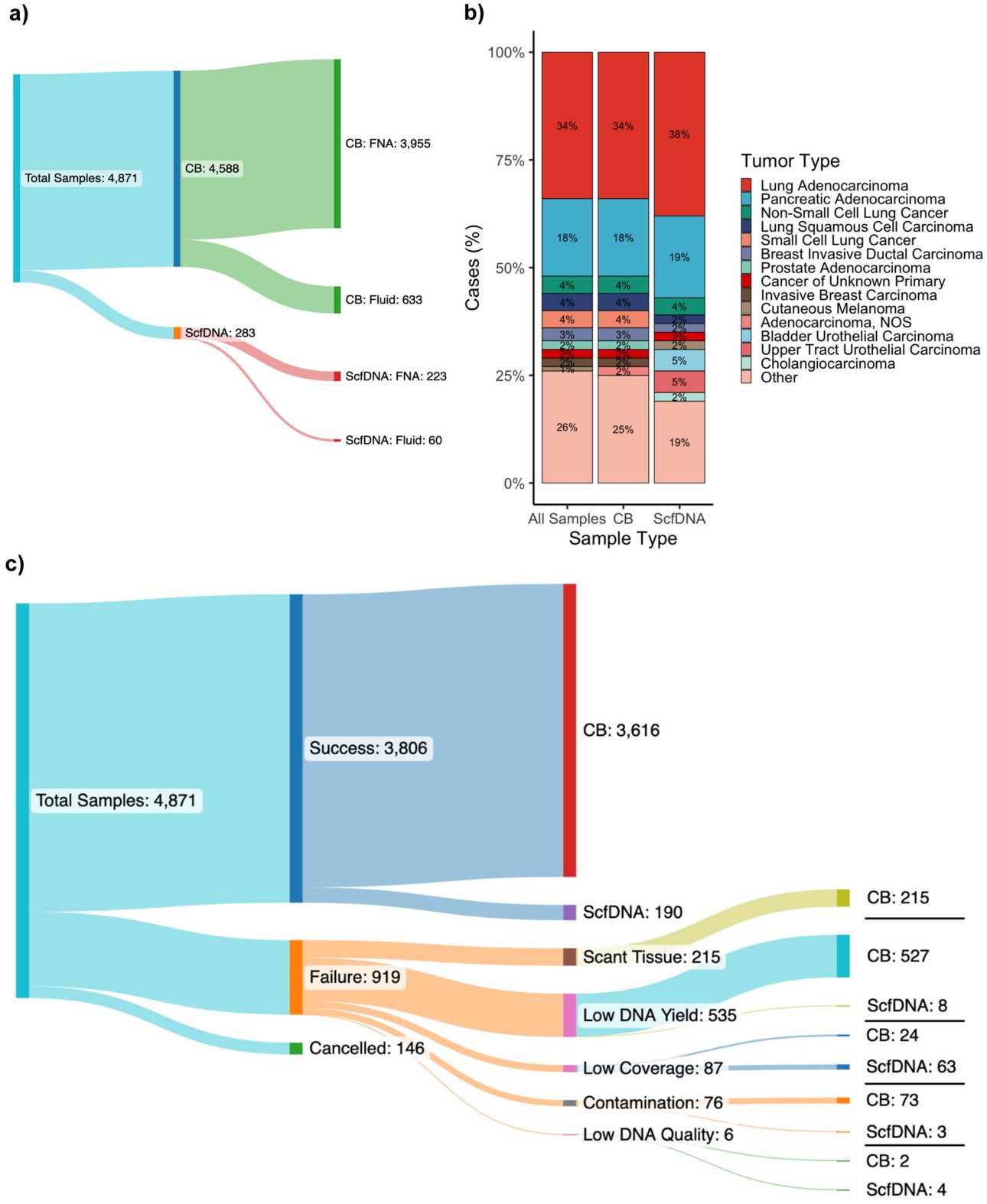

**Fig. 1 | Overview of clinical cytology samples profiled by MSK-IMPACT. a** The composition of the cytology cohort by sample preparation type, either cellblock (CB) or supernatant cfDNA (ScfDNA), and their respective collection method of either fine needle aspiration (FNA) or cytology fluid (e.g. pleural fluids, bronchial washes, ascites fluid, etc.). **b** The distribution of cancer types profiled by MSK-IMPACT clinically from cytology samples. The most common cancer types are at the top and are ordered in descending order for specific cancer types. Represented are all cytology samples and by sample type. **c** The distribution of testing outcome and sample type of cytology samples. In cytology cases that failed testing for MSK-IMPACT the cause of failure is further broken down with the corresponding number of sample types for each reason.

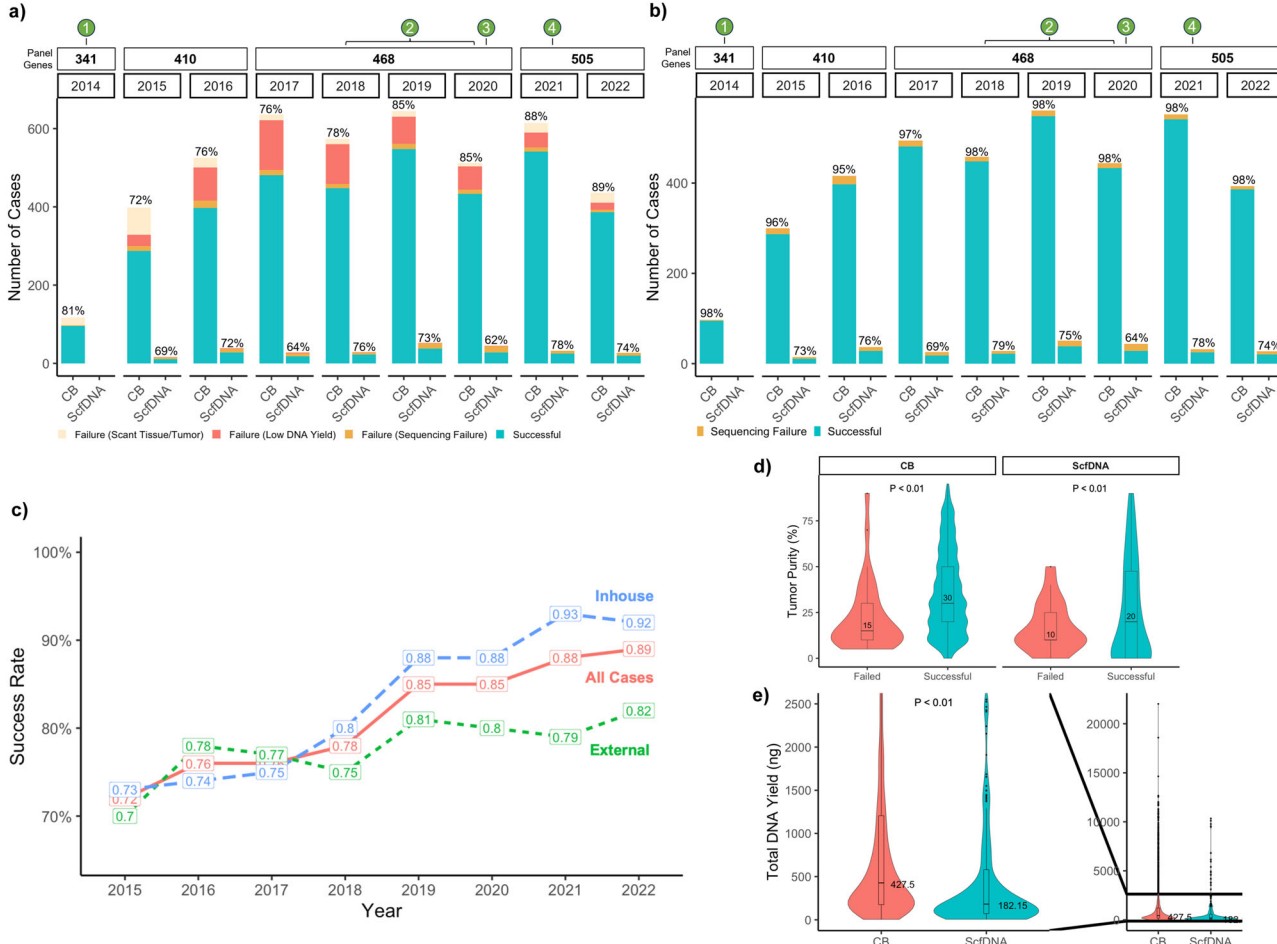

**Fig. 2 | Determinants of successful mutational profiling by MSK-IMPACT in cytology samples. a** Success rates of MSK-IMPACT testing on cytology samples by year and sample type. The colors denote successfully tested samples, samples failed due to scant/low tumor tissue, failed due to DNA content below sequencing thresholds, and all other failures (e.g., low DNA quality, contamination, etc.). Throughout the study period the panel genes increased in number and are denoted in the top bar by the number of panel genes included for that year. Various key optimization efforts for cytology samples were introduced in the clinical workflow at the corresponding timepoints including the use of a modified HistoGel cell-block preparation (1), improved bead-extraction procedures (2), dual-indexed libraries (3), and decreased lower DNA input for sequencing (4). **b** Success rates charted similar to that seen in (2a) but including only samples that were deemed to have adequate tumor for sequencing. The success rates therefore indicate the success rate of cytology sampled deemed to have adequate tumor on manual review as opposed to the overall testing success rate of a cytology sample received for testing. **c** Overall sequencing success rates on MSK-IMPACT for cytology samples charted by year and stratified by samples processed at an external laboratory (External), samples processed at the study institution (Inhouse), and all cytology cases (All Cases). Two-sided Chi-squared test revealed a significant difference in success rates between External and Inhouse samples in the years 2019 ($p = 0.0087$), 2021 ($p = 0.0001$), and 2022 ($p = 0.0004$). **d** The distribution of tumor purity for cytology samples by outcome of MSK-IMPACT testing for CB ($n = 4457$; $p = 0.00039$) and ScfDNA ($n = 268$; $p = 0.63$) samples. **e** Comparisons of total extracted DNA yields for extracted CB ($n = 4457$) and ScfDNA ($n = 268$) cytology samples ($p = 2.2 \times 10^{-16}$). Group comparisons for continuous data (**d**, **e**) were performed with a two-tailed Mann–Whitney test set at a $p < 0.01$. All boxplots show the median (center line with value) and 25th and 75th percentiles (bounding box) along with the 1.5 interquartile range (whiskers). Source data are provided as a Source Data file.

(246/4725) (Fig. 3c). Excluding those that failed due to very low coverage (<50×), the overall rate was 4.8% (227/4725), with a significantly higher rate for CB samples compared to ScfDNA, at 4.7% (226/4725) and 0.3% (1/4725) respectively. Notably, in the context of optimal coverage (>200×), no ScfDNA samples exhibited clinically significant sample contamination (Fig. 3d). Also, no contamination was identified, even for samples with low coverage, following the implementation of dual indexing. By contrast, CB samples showed variable and significantly higher levels of contamination (range: 2–32%), which remained present despite adequate coverage and after implementation of dual indexing. Among samples with optimal coverage, 4% ($n = 189$) of the CB samples exhibited contamination rates above 2%. In 34% ($n = 65$) of these samples, for which sufficient material for re-extraction and Short Tandem Repeat (STR) analysis was available, contamination could be tracked to foreign tissue material embedded in the tissue blocks. Representative cases are included in Supplementary Fig. 1.

## Biomarker/mutation identification and therapeutic actionability

A total of 30,149 somatic alterations were detected across the 3806 successfully sequenced cases. Of these, 93.8% of cases (3570/3806) harbored at least 1 somatic alteration, including 3394 (93.9%) of CB and 176 (92.6%) of the ScfDNA samples. No significant differences in sample coverage were observed between samples with and those without alterations (CB: $p = 0.19$; ScfDNA: $p = 0.91$, Supplementary Fig. 2). However, tumor purity estimations were significantly lower for the subset without alterations, with a median tumor content of 10% vs 30% for those with detected alterations.

When stratified, the median number of alterations was similar for both sample types, 9 for CB (range: 1–170; 95% CI: 11.9–12.7) and 10 for ScfDNA (range: 1–63; 95% CI: 8.5–11.9). The average TMB was 7.64 mutations/Mb and 7.43 mutations/Mb for CB and ScfDNA samples, respectively. Overall, the mutational profiles recapitulated the

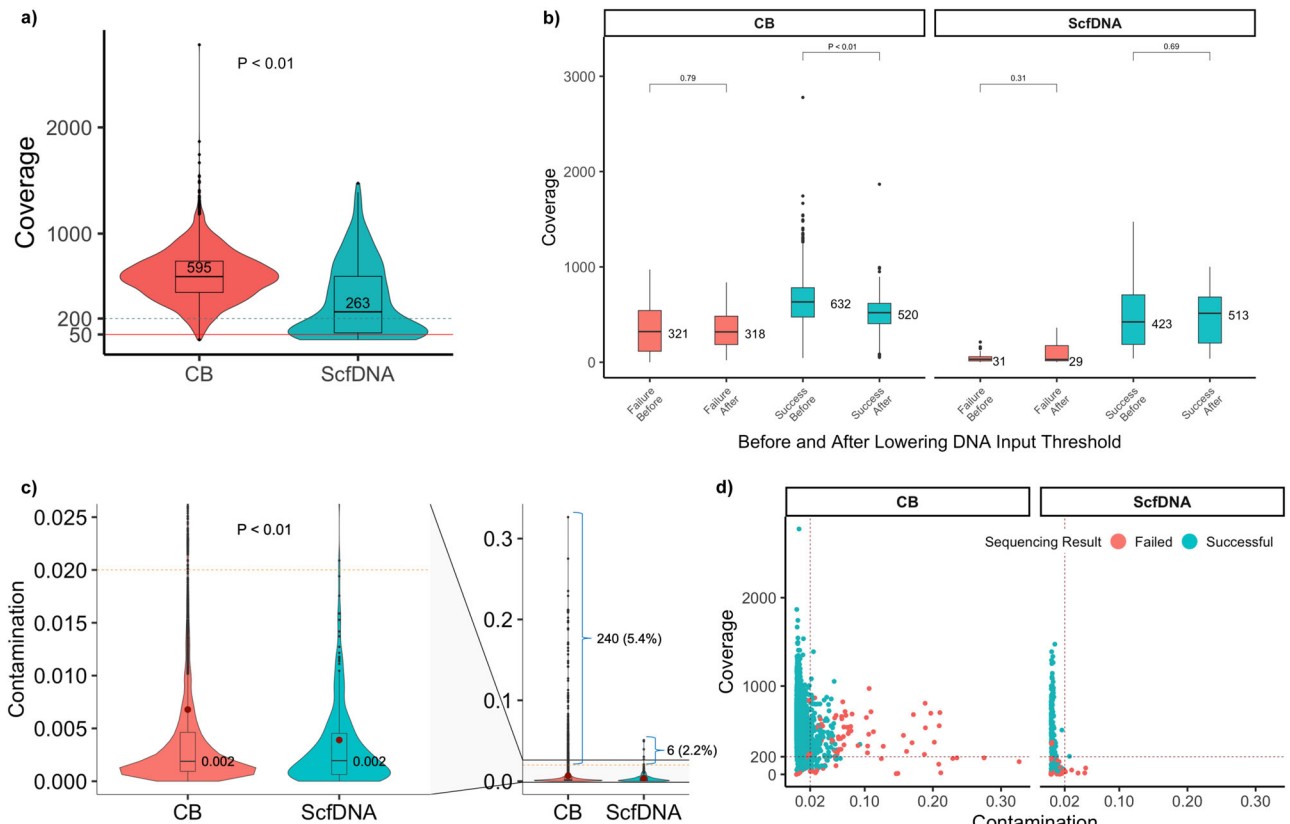

**Fig. 3 | Sequencing quality metrics of profiled cytology samples. a** Sample coverage distribution by sequenced CB ($n = 3707$) and ScfDNA ($n = 259$) cytology samples ($p = 2.2 \times 10^{-16}$). The dashed horizontal line indicates the 200× coverage mark in which samples are deemed to have low coverage with concerns for false negative assessment. The solid red line indicates the 50× mark for which samples below this mark are highly considered to be failed due to low coverage. **b** The distribution of total sample coverage before and after the lowered DNA threshold for sequencing by MSK-IMPACT compared in failed CB samples ($n = 841$, $p = 0.79$), successful CB samples ($n = 3616$, $p = 2.2 \times 10^{-16}$), failed ScfDNA samples ($n = 78$, $p = 0.31$), and successful ScfDNA samples ($n = 190$, $p = 0.69$). **c** The distribution of non-patient DNA contamination rates and mean (red dot), determined by comparing homozygous SNP sites between the sequenced tumor and matched patient normal sample for CB ($n = 4510$) and ScfDNA ($n = 259$) samples ($p = 0.00015$). The dashed horizontal line indicates a contamination rate of 0.02 for which samples

with a higher rate are considered to have a concern for non-patient DNA contamination. The number and percentage of samples that exceed this threshold are shown adjacent to the representative bracket for each sample type. **d** The distribution of contamination rates between sample preparation methods charted by sample coverage. Samples are colored based on sequencing results. The dashed vertical line indicates the threshold contamination rate of 0.02 and the dashed horizontal line denotes the threshold for adequate coverage (200×). Samples in the top-right quadrant indicate a high contamination rate in the face of adequate coverage, whereas samples in the bottom-right had low coverage that may falsely elevate the contamination rate. Group comparisons for continuous data (**a–c**) were performed with a two-sided Mann-Whitney test set at a $p < 0.01$. All boxplots show the median (center line with value) and 25th and 75th percentiles (bounding box) along with the 1.5 interquartile range (whiskers). Source data are provided as a Source Data file.

expected landscape and frequency of driver and common alterations for the tumor type. Stratified by level of actionability, 65% ($n = 2487$) had at least one targetable alteration as defined by the presence of an OncoKB level 1, 2, 3A, or 3B alteration and 2% ($n = 93$) had a standard care resistance mutation (OncoKB level R1). The highest frequency of level 1 OncoKB alterations was observed in thyroid, breast, non-small cell lung (NSCLC), and bladder cancer patients at 58%, 58%, 45%, and 29% respectively. For resistance mutations, 87 CB and 6 ScfDNA samples identified an OncoKB level R1 alteration. To ensure that significant alterations were being identified at similar rates to non-cytology samples, results were compared to those published in the AACR GENIE cohort. Across the different histologic tumor types, similar rates of OncoKB alterations were identified (Fig. 4a). Representative oncoplots of the most frequent, clinically actionable alterations detected in NSCLC, Bladder Cancer, and Breast Cancer (most common tumor types in our cohort) are presented in Fig. 4b, c which demonstrate the expected distributions across both CB and ScfDNA. OncoKB level 1 alterations were commonly seen in *EGFR, KRAS, PIK3CA*, and *ERBB2* genes. *ALK, BRAF, RET*, and *ROS1* level 1 alterations were also seen at lower frequencies.

## Comparison with surgical core biopsies/resections

To further assess the general performance of cytology samples, we identified 526 cases (CB: $n = 482$; ScfDNA: $n = 44$) of patients who had a corresponding surgical sample of the same tumor assessed by MSK-IMPACT. While the same tumor was profiled across each surgical:cytology pair, it should be noted that there were variations in the time of collection across their treatment course as the samples were profiled clinically. Thus, many of the cytologic samples were collected at the time of disease progression or development of resistance. Overall, cytology samples delivered high performance with similar sequencing metrics compared to their surgical biopsy/excision counterparts. The average coverage, while high for both sample types and retaining a similar technical sensitivity, was slightly lower in cytology samples (584× vs 628×), reaching statistical significance ($p = 0.00028$).

Comparing detected alterations, a large proportion of the cytology samples identified all the alterations detected on the corresponding surgical sample, 266 (55%) CB (Fig. 5a) and 19 (43%) ScfDNA samples (Fig. 5e). The median variant allele frequency (VAF)'s for shared alterations were slightly higher for cytology samples, compared

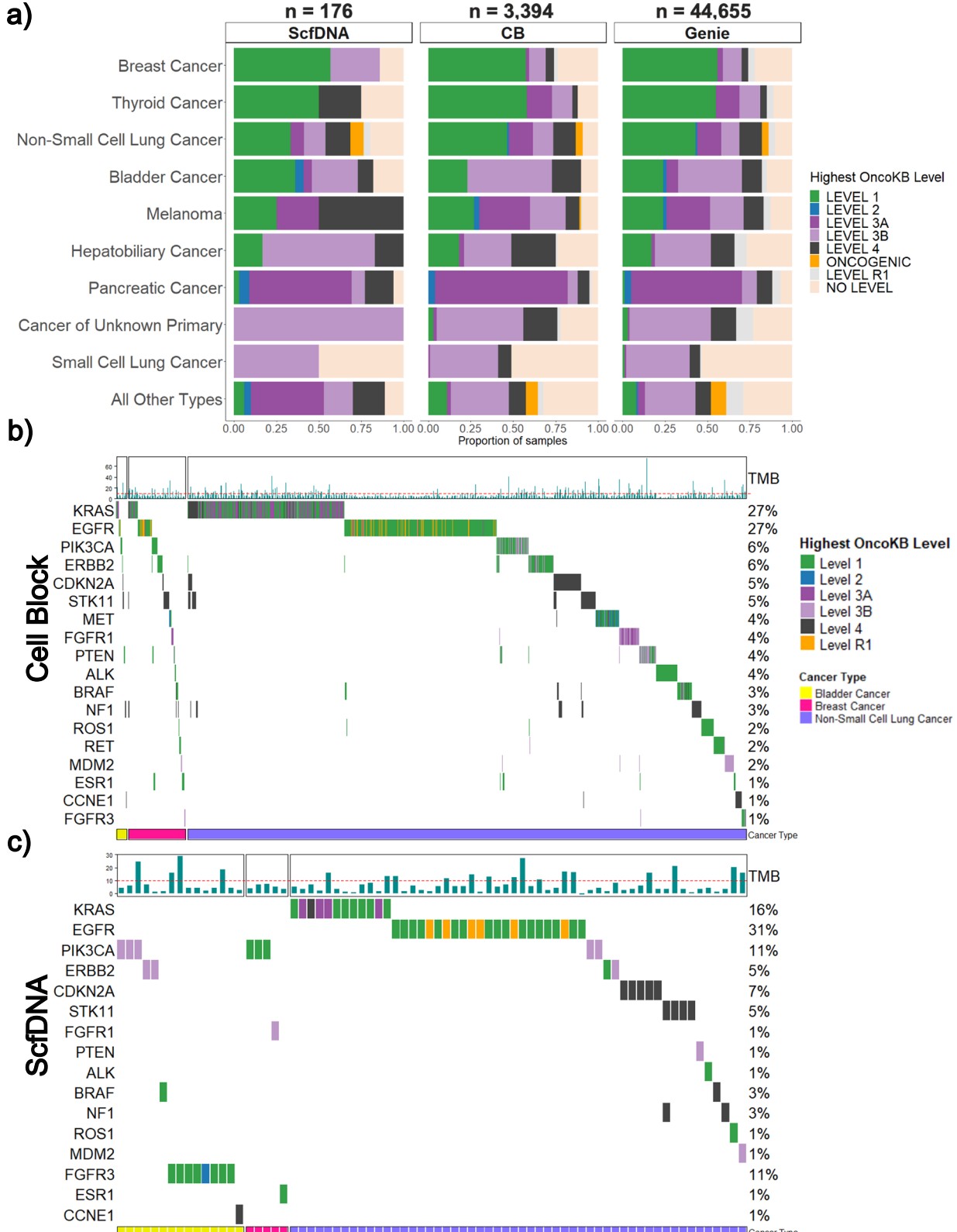

**Fig. 4 | Comparison of mutation calls between cytology cellblock and supernatant cfDNA samples. a** The proportion of significant genomic alterations identified by tumor type in ScfDNA and CB cytology samples compared to the pan-cancer GENIE cohort (non-cytology). The various colours indicate the highest OncoKB level and oncogenicity associated with the alteration identified. Comparative oncoprints of significant genomic alterations in three common tumor types profiled (bladder cancer, breast cancer, and non-small cell lung cancer) for samples with reported alterations identified from (**b**) CB and (**c**) ScfDNA samples. The significance of genomic alterations is coded by the corresponding OncoKB level. Sample level tumor mutational burden (TMB, mutations per megabase) is provided for each corresponding sample at the top with the horizontal dashed line indicating 10 mutations/megabase, for which samples with a higher number are TMB-High.

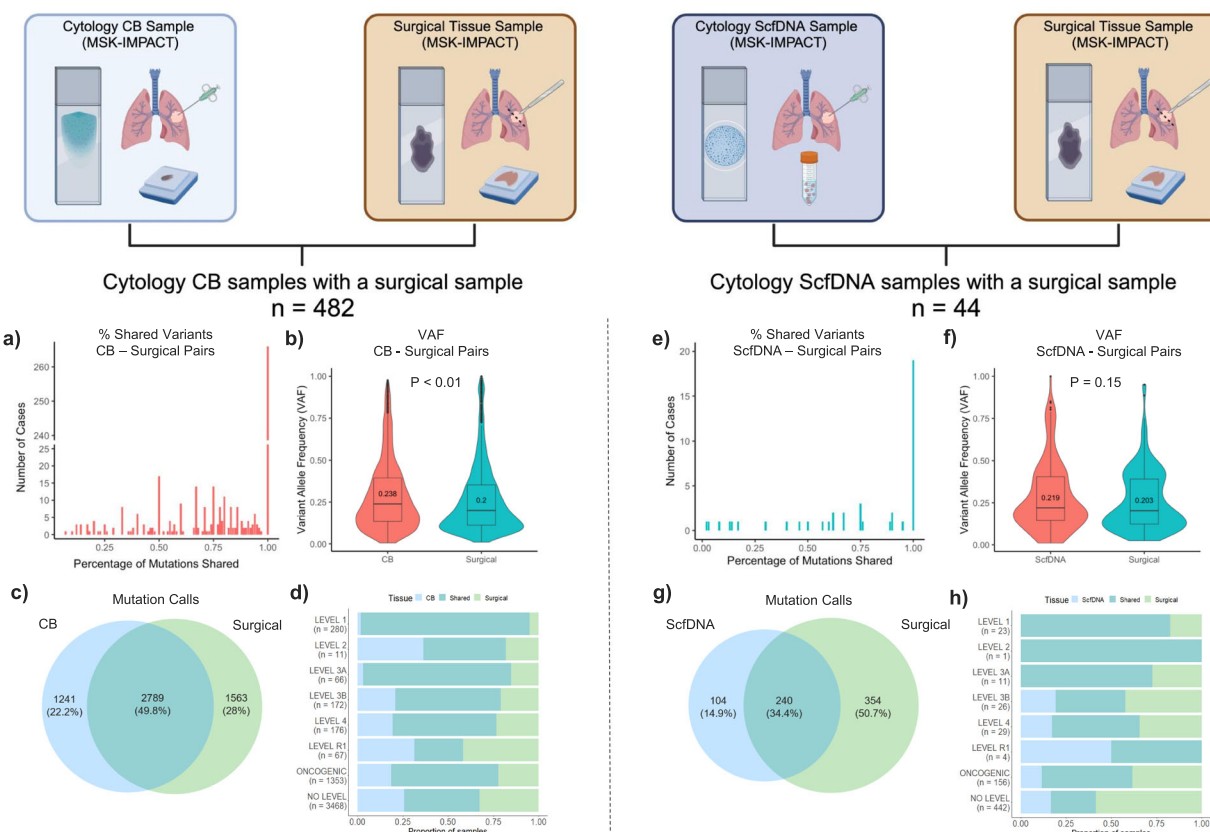

**Fig. 5 | Comparison of mutation calls by MSK-IMPACT between matched cytology and surgical samples.** Cytology samples with a corresponding surgical sample (e.g., core biopsy, resection, etc.) of the same patient tumor profiled by MSK-IMPACT (*n* = 526) for comparison were identified. A total of 482 cytology CB samples profiled by MSK-IMPACT had a corresponding surgical sample profiled by MSK-IMPACT. **a** The number of cases tallied by the proportion of genomic alterations identified on the surgical sample that was also identified on the corresponding cytology CB. **b** The distribution of variant allele frequency (VAF) of shared mutations identified on both cytology CB samples and corresponding surgical samples ($p = 4.9 \times 10^{-11}$) of the same patient tumor. **c** Venn diagram representing the total mutation calls in cytology CB samples only (left), their corresponding surgical sample only (right), and those found in both (middle). **d** The proportion of significant alterations identified in the cytology CB and corresponding pairs analyzed by if the mutation was identified exclusively in the cytology CB, corresponding surgical sample, or if it was seen in both (shared). The same analysis performed for ScfDNA samples (*n* = 44) with the proportion of genomic alterations identified (**e**), VAF distribution of shared alterations (**f**), venn diagram of mutation calls (**g**), and proportion of significant alterations identified by tissue sample and those seen in both (**h**). The *p* values were assessed as group comparisons for continuous data with a two-tailed Mann–Whitney test set at a *p* < 0.01. All boxplots show the median (center line) and 25th and 75th percentiles (bounding box) along with the 1.5 interquartile range (whiskers). Illustrations created with BioRender.com (BioRender.com/n48n733).

to the surgical pair in both CB and ScfDNA samples (CB: $p = 4.9 \times 10^{-11}$; ScfDNA: $p = 0.13$) (Fig. 5b, f).

In all, a total of 5593 mutations were identified in the CB:surgical paired set, of which 2789 events (49.8%) were shared (Fig. 5c). For the ScfDNA:surgical paired set, 692 mutations were detected with slightly lower overlap (34.8%; Fig. 5g). Importantly, when alterations were stratified by level of actionability, the overwhelming majority of driver alterations with OncoKB Level 1 actionability were shared events, at 93% and 83% for the CB and ScfDNA sets, respectively. Non-detection of OncoKB Level 1 alterations in the surgical or the cytology sample was related to low coverage or low tumor content in all cases. Events categorized as Level R1 or No level showed the lowest overlap, with 27% and 42% shared events, respectively. It should be noted that mutations categorized in the No Level category (variants of unknown significance) constituted the largest proportion of all mutations (62%; 3468/5593) and were responsible for most of the discrepancies encountered. Non-detected variants were equally distributed across surgical and cytologic samples, likely reflecting the heterogenous nature of the alterations (differential passenger events, acquisition of additional mutations in the time interval of the two samples), rather than false negativity in testing. Discordance in mutations categorized as Level R1 categories were primarily

attributed to acquisition of resistance mechanisms in later samples or to the heterogeneous presence of resistance alterations across tumors. Further details are provided in Fig. 5d, h.

Review of contamination check data for surgical samples revealed that <1% of surgical samples had clinically relevant contamination (0.81%; 5/619; Supplementary Fig. 3a). Of note, all 6 surgical samples with contamination were minute biopsy samples with low tumor purity (Supplementary Fig. 3b) with contamination below 4%.

## Comparison of successful CB preparations with corresponding ScfDNA

Among CB cytology samples with adequate tumor and successful sequencing, 24 had the corresponding ScfDNA samples tested to allow direct comparisons. Both DNA concentration and sequencing coverages were significantly lower for the ScfDNA samples. Total DNA yields averaged 505 ng (range: 76–5453) and 1200 ng (range: 83.4–3340) for ScfDNA and CB preparations, respectively. Accordingly, ScfDNA had resulting lower sequencing coverage averaging 387× (range: 3×–1335×) compared to 669× (range: 74×–1193×) for the corresponding CB. Of the 24 ScfDNA samples, 7 (27%) failed sequencing due to low sample coverage. Detection of clinically relevant alterations and the VAF was the same across both sample preparations based on a comparison of successfully sequenced sets.

## Discussion

Comprehensive testing by NGS is becoming a common approach for the upfront assessment of a broad range of genetic biomarkers that are pivotal for diagnostic, prognostic, and therapeutic decisions in cancer patients. While ideally, molecular testing is greatly facilitated when large tumor samples are available (i.e. resections or excisional biopsies), the reality of clinical practice is that a very large proportion of testing must be performed on scant material obtained through minimally invasive procedures. Historically, this has presented distinct challenges, prompting the adoption of alternate approaches, such as liquid biopsies, which attempt to circumvent tumoral cell assessment altogether. At present, while arguments can be made for the superiority or inferiority of each modality over another, cytologic samples stand as the one middle approach that unites the most desirable attributes of both worlds. Namely, they retain the key morphologic correlates required for tumor diagnosis, while still sparing the patient from the more invasive procedures. One fact remains constant, however, which is that small samples require very high optimization of the entire process to maximize the genomic yield.

In this study, we have outlined our institutional approach and longitudinal experience in the comprehensive profiling of cytology samples in routine clinical care. Our experience of sequencing 4871 prospective clinical samples demonstrates that molecular testing can be performed on routinely procured cytology samples with high success rates, similar to surgical samples with various optimization strategies. Proportions of clinically actionable genomic alterations, specifically OncoKB Level 1–3B, as well as R1 alterations, recapitulated the expected patterns across all tumor types when compared to those published in AACR GENIE cohort. For immediately actionable alterations (OncoKB Level 1), the concordance of cytology to corresponding surgical samples from the same patient were very high (93%). Notably, rescue ScfDNA from supernatant CytoLyt fluid material, utilized for our internal cases, proved highly valuable and enabled the detection of a Level 1 alteration in 83% of the successfully sequenced cases.

Our review of data compiled across 8 years, highlighted the central roles of optimized sample handling and processing. In our hands, 2 critical early steps enabled higher DNA recovery which, consequently, promoted increased utilization of cytologic material for molecular testing. The first was the optimization of cell block preparation, which incorporated pretreatment of pelleted cells with 95% ethanol before addition of HistoGel[12,13]. This enhanced the density of cell pellets to deliver a higher amount of cellular material in fewer sections of the paraffin block. The second was the transition to mineral oil deparaffinization which markedly reduced tube transfers, centrifugation, and decanting steps, all key vulnerabilities responsible for major nucleic acid losses in the processing of scant FFPE material[14–16]. It should be noted that, with the implementation of mineral oil extraction, requests for testing on cytologic material vs needle biopsies markedly increased at our institution. Details of this transition have been previously published by our group[3,16]. In particular, among lung cancer patients undergoing endobronchial ultrasound transbronchial needle aspiration (EBUS-TBNA), this change alone significantly improved sequencing success rates from 76.3% to 93%. Moreover, these success rates corresponded to NGS testing that was performed after standard rapid testing for EGFR on the same samples[18], further supporting the high suitability and sufficiency of the DNA recovered.

An important, and often underreported, consideration in molecular testing of cytology samples are the diagnostic challenges and inaccuracies that may arise from sample cross-contamination. While sample-to-sample contamination may happen across any point, highly vulnerable points lie in processes that involve batching and pooling of multiple samples in a single run. In particular, established histopathology practices of tissue processing (i.e., carry over from microtome blades, common water baths, pooled tissue processors, etc.), pose distinct risks for contamination for small tissue samples as processes are primarily optimized to enhance microscopic diagnostic analysis but not downstream molecular applications. Common holding of numerous specimens in single chambers in automated tissue processors and the use of common equipment for embedding, cutting, and tissue mounting all increase the potential for low-level cross-contamination. While this may remain inconsequential for morphologic assessment or the molecular analysis of large tissue samples, this can distinctly impact small samples where similar contamination levels become proportionally higher. Cytology samples may be even more vulnerable due to the processing of cell blocks with paper wrapping and HistoGels, which may promote the trapping of cellular impurities from other samples. Indeed, in our analysis of cytologic samples, contamination was significantly higher across CB samples compared to all other samples. This held true when analyzing samples with adequate coverage (>200×) with 4% of CB samples exhibiting contamination. ScfDNA samples, by contrast, which are processed individually in a closed system and not batched, had negligible levels with contamination patterns exclusively associated with sequencing failures or borderline coverage and more likely related to artifact rather than true contamination. Despite the presence of higher contamination levels in cell blocks, the overall rate was low (4.8%) among successfully sequenced samples, which encompassed samples procured and processed in numerous laboratories across the county. These rates are in keeping with sequencing data on surgical samples published by Sehn et al.[19] but are significantly lower to what is reported by the American Society of Cytopathology (ASC) Clinical Practice Committee/Workgroup for Cross-Contamination in a recent survey for general cytopathology practice, quoting rates as high as 56% for cell-block preparations[20]. This high rate may be related to the reporting of contamination per case, affecting some but not all unstained sections and which may not be high enough to be detectable in the sequencing of DNA recovered from a set of several slides. Importantly, while contamination was detectable in several cases in our cohort, most were sufficiently low in comparison to the overall tumor content of the sample, allowing informed filtering of low–variant allele fraction events without compromising all mutation calling. In all, only 1.3% of the samples were failed due to contamination, while others could be reported with modification. Within the molecular laboratory, a notable source of cross-contamination may arise from index-hopping during multiplexing. This, however, is generally lower level (well below 2%) and more prone to affect higher sensitivity applications. Nonetheless, in the process of improvement for our MSK-IMPACT assay we have incorporated several strategies to mitigate this phenomenon, including optimization of PCR conditions and the implementation of dual indexing to facilitate the removal of misaligned reads. These finetuning steps facilitated our decision to decrease the minimum assay input requirements which markedly reduced our failure category due to insufficient DNA. No significant changes in coverage or contamination rates were seen with this change.

Finally, a pivotal component of our optimization process was the implementation of testing ScfDNA recovered from liquid cytology preparations. While, generally, this sample type was not submitted if the cell block was deemed suitable for testing, it became an important rescue sample to avoid re-biopsy procedures. The use of this material also relieved some of the challenges in triaging very small biopsy samples for other ancillary studies. In all, while the success rate of the ScfDNA samples was approximately 71%, which is below what is seen across tissue biopsies and CB, these samples were specifically tested after the corresponding cytologic material was deemed unsuitable, thus boosting the overall success for the individual aspirate procedures by approximately 3%. An important observation, gathered from the comparison of ScfDNA and corresponding cell blocks, is that the VAF's of detected alterations were similar for both preparations, supporting that the assessment of the block or cytoprep represents a suitable surrogate for estimating the proportion of tumor-derived

DNA that may be present in the ScfDNA sample. Additionally, given the high integrity of the DNA in these non-formalinized samples, lower DNA inputs still delivered excellent results, provided that the tumor proportion was suitable. Confirmatory testing with higher sensitivity methods may also be implemented for low tumor samples, without concerns for false positivity due to artifacts imparted by formalin fixation.

In conclusion, this study confirms that the routine use of cytologic samples for molecular testing constitutes a robust approach that can deliver the same results as larger biopsy samples, provided that adequate tumor content is present. Process optimization and the implementation of robust quality control processes, including contamination checks are pivotal to maximizing the yield and utility of these samples. A reassessment of how tissue blocks are processed and prepared would be an important aspect of cytology practice as a whole, to include specialized instrumentation for processing small samples without risk of cross-contamination. ScfDNA recovered from supernatants is an invaluable source of tumor-derived DNA which circumvents the processing where most contamination is bound to happen in current practice, and while failure rates due to limited nucleic acid recovery are higher than tissue blocks, their use could rescue the majority of cases where the high tumor is identified but FFPE material is insufficient for sequencing.

## Methods

### Patient consent and cohort selection

This study complies with all relevant ethical regulations. All samples in this study were collected with informed consent from the patients for routine prospective clinical genomic analyses. MSK-IMPACT[TM] testing was ordered by the treating physician to identify clinically significant genomic alterations for the clinical care of patients with cancer. Informed consent for the molecular profiling of patient tumor was obtained under protocol NCT01775072 (Tumor Genomic Profiling in Patients Evaluated for Targeted Cancer Therapy) along with a match normal plasma sample for paired tumor/normal testing. The protocol was approved by the Institutional Review Board at Memorial Sloan Kettering Cancer Center and written consent was obtained from all patients.

The prospectively maintained database of samples submitted for sequencing using our institution's large-panel NGS assay (MSK-IMPACT[TM]) between the years 2014 and August 2022 was queried to identify all cytology samples; this included all requests on samples deemed to be malignant by morphologic assessment prior to their assessment of suitability for sequencing. Basic demographic data (age and sex as self-reported) and any existing pre-analytic information, including the type of preparation, tissue source, tumor type, tumor content, and DNA yield, were collected. Samples in this study were not chosen based on sex or gender but were included if the sample was sequenced clinically. All sequencing data, encompassing QC metrics, sequencing coverage, somatic variants identified, and VAF were gathered, as well as sequencing qualification (pass, fail, reason for failure) as established at the time of clinical signout. When available, the same sequencing metrics and information described above were also collected for corresponding biopsy and resection samples from the same patient tumor to compare results side-by-side. All samples were collected with informed consent, and testing performed in our CLIA-certified laboratory.

### Cytology sample processing

Cytologic samples were received as formalin-fixed paraffin-embedded (FFPE) tissue sections from cell block (CB) preparations or as supernatants. Samples were either collected in CytoLyt fixative (Hologic, Malborough, MA, USA) or in 10% neutral buffered formalin fixative and were paraffin-embedded (FFPE). CB preparation for MSKCC procured samples followed a modified HistoGel-Based Cell Block Preparation

Method that utilizes the addition of 95% ethanol before the standard HistoGel Cell Block preparation[12,13]. Procedural details of externally procured samples (cases submitted for review at MSKCC for diagnosis confirmation and IMPACT testing) were not available. For each case, 20 unstained sections (5 μm thick) were submitted mounted on glass slides, along with a hematoxylin and eosin–stained (H&E) section to assess adequacy and tumor fraction. Macro-dissection was performed to enrich for tumor, when possible and necessary, aiming for >50% tumor cell content. Samples were rejected/failed if the tumor proportion was <10% and the sample was not amenable to manual enrichment.

For MSKCC samples that were collected in CytoLyt fluid, residual material was saved after the ThinPrep® and CB were prepared. The corresponding ThinPrep® slide was assessed as a surrogate for tumor presence and content. If tumor cells were present at ≥10% based on visual inspection, the supernatant was considered suitable, and DNA was extracted. Further details of the processing of supernatants for ScfDNA extraction are described in a previous publication[16].

### Extraction procedures and DNA quantitation

FFPE material was deparaffinized using Citrasolv (2014 to March 2016) or mineral oil (March 2016 to 2022), and DNA was extracted using the Chemagic STAR DNATissue-10 Kit (Perkin Elmer, Waltham, MA) with the magnetic-bead method automated on a Chemagic STAR Standard Solutions Workstation (Hamilton, Bonaduz, GR, Switzerland), following manufacturer's protocols. DNA from supernatants was extracted using the same kit and automated system, eliminating deparaffinization and overnight lysis incubation at 56 °C and, instead, 1 hour lysis incubation was used (56 °C).

Extracted DNA was eluted and quantified using a Qubit DNA high-sensitivity assay kit (Life Technologies, Carlsbad, CA). FFPE samples with DNA concentration of <0.9 ng/uL were deemed insufficient for further testing until 09/2021 when the threshold was lowered to 0.54 ng/uL to proceed with sequencing, which translates to minimal total inputs of 50 and 30 ng, respectively, based on maximal volume inputs for the assay of 55ul. The change in the minimal input cutoffs was informed by the ongoing review of assay performance. As reference, based on existing validation data for MSK-IMPACT, the estimated number of target DNA molecules/copies per nanogram of DNA across FFPE samples of variable quality averages 238 ng (range 14–489 ng; 95% CI: 220.5–255.5), as established by digital droplet PCR assays. Therefore, for samples at the lowest DNA input of 30 ng, we expect to query at least 6615 target molecules in 95% of cases and allow high performance of the assay. ScfDNA samples were sequenced below these thresholds, aiming to spare the patient from a future biopsy and to further evaluate performance characteristics.

### Next generation sequencing

DNA was sheared and processed (along with matched DNA from blood as normal control) to generate bar-coded libraries which were pooled and subjected to targeted capture using custom-designed probes[17]. All samples in this study underwent testing by MSK-IMPACT[TM], targeting all coding regions of up to 505 genes, select introns and over 1000 custom intergenic and intronic regions throughout the genome (centered on common SNPs). Updates to the panel sequentially increased the number of genes captured from 341 (year 2014), 410 (years 2015–2016), 468 (years 2017–2020), to 505 (years 2021–2022). Captured DNA fragments were sequenced on an Illumina HiSeq2500 or NovaSeq 6000 system, before being submitted to the bioinformatics analysis pipeline for calling of somatic alterations. Clinical actionability and treatment associations of the genomic alterations detected were assessed and annotated using OncoKB[21] MSK's precision oncology knowledge base. Levels of evidence are assigned to each alteration based upon therapeutic levels of evidence specific to the tumor type profiled, including alterations predictive of resistance to therapy.

Results were compared to existing, highly curated, MSK-IMPACT clinical data from corresponding solid tumors (non-cytology), which have been previously published as part of the American Association for Cancer Research (AACR) Project Genomics Evidence Neoplasia Information Exchange (GENIE) Pan-Cancer Cohort[22]. In all, this encompassed results from 44,655 unique tumors, with a direct focus on the clinical actionability of the alterations detected.

Tumor mutational burden (TMB) was calculated for each sample by dividing the total number of non-synonymous mutations, including driver mutations in oncogenes, by the total genomic target region of the respective MSK-IMPACT panel version. Calculated values were reported as mutations per megabase (Mb).

Samples were deemed clinically successful if they passed all quality control metrics defined for our assay (e.g., adequate tumor quantity, coverage, base quality, etc.) and were formally reviewed by a board-certified molecular pathologist before the report was released clinically.

### Next-generation sequencing quality metrics and contamination assessment

For QC purposes, in addition to standard NGS quality metrics, assessment for potential sample contamination was a critical component of our assessment. Our established analysis pipelines compute pairwise genotype concordance across all SNP sites included in the panel. This unique genotype analysis, enabled by our paired tumor:normal sequencing approach, allows us to identify potential sample swaps and contamination, either due to the presence of DNA from another individual or contamination among different barcode adapters, which could lead to erroneous mutation calling. Contamination levels are defined by the analysis of SNP sites at which the patient is homozygous (based on normal control profile). Because a homozygous site is defined by 2 identical alleles at the particular genetic locus, any allelic discrepancy where the variant is not expected indicates contamination. A cutoff of ≥2% is used to denote clinically significant contamination (the threshold for mutation calling). Samples with a contamination rate higher than 2% were evaluated in the context of the tumor content and mutation profile; low-level contamination in samples with very high tumor content, remained partially evaluable by filtering variants within the range of contamination.

### Sample type comparisons and concordance analysis

To further assess the performance of cytologic samples compared to larger tissue samples, existing MSK-IMPACT sequencing data from corresponding tissue core biopsies or subsequent resection samples from the same tumor were obtained. Once matched, metrics including sequencing coverage, genomic alterations identified, mutation VAFs, and OncoKB levels were compared to the cytologic counterpart. In a subset of cytology cases, results from CB deemed adequate for testing were also compared to the corresponding ScfDNA. This data was analyzed separately to avoid duplication.

### Statistical analysis

Statistical analysis for group comparisons of continuous data were performed using a two-tailed Mann–Whitney test. A Pearson's Chi-squared test was performed for comparing three or more categorical groups. A Fischer's exact test was performed for comparing two categorical groups. Statistical significance was set at $p < 0.05$. Cases with missing values were removed from the analyses, and only complete cases were considered. All statistics and graphical representations were performed using R project. No statistical method was used to predetermine sample size.

### Reporting summary

Further information on research design is available in the Nature Portfolio Reporting Summary linked to this article.

## Data availability

The raw sequencing data for MSK-IMPACT analysis is protected and cannot be broadly available due to privacy laws. Patient consent to deposit the raw sequencing data was not obtained. Data supporting the analysis for all samples, including all genomic results, in this study, are included in this published article, Supplementary Information, and Source Data files. Unique sample IDs of the study samples are provided in the Source Data files under Supplementary Fig. 2 and available for query on cBioportal: https://www.cbioportal.org/. AACR GENIE sample data used in this study is available at https://genie.cbioportal.org/?continue. The AACR GENIE sample data can be accessed after proper registration as a first-time user at https://docs.google.com/forms/d/e/1FAIpQLScwlJ9WRmAGZ08CCg8wYo8l8bcUmsAzJ09i1MKjBNtb_dLqIw/viewform. Source data are provided with this paper.

## Code availability

Analysis was performed in R (version 4.0.3). Analysis code for in-house developed pipeline modules is made available on Github (https://github.com/rhshah/IMPACT-Pipeline).

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

## Acknowledgements

This research was funded in part through the NIH/NCI Cancer Center Support Grant P30 CA008748.

## Author contributions

All authors of this article declare that we qualify for authorship for this manuscript. Each author has participated sufficiently in the work and takes public responsibility for appropriate portions of the content of this article. D.K. and M.A. carried out the background research and drafted the manuscript. D.K. and M.A. conceived the idea for the manuscript, its design, and coordination. D.K., C.V., A.Y., and S.N. generated and interpreted the data. D.K. and C.V. performed the statistical analysis for this study. D.K., C.V., S.Y., S.N., K.N., R.F., N.R., I.R., J.C., A.Y., A.R.B., M.B., M.L., O.L., and M.A. reviewed the manuscript critically for important intellectual content. All authors read and approved the final manuscript.

## Competing interests

C.V. reports intellectual property rights and equity interest in Paige.AI, Inc. A.R.B. has ownership/equity interests in Johnson and Johnson. M.B. has received advisory or consulting fees from AstraZeneca, Eli Lilly and Company, and PetDx, Inc. M.L. has received advisory or consulting fees from Takeda Oncology, Janssen Pharmaceuticals, AstraZeneca, ADC Therapeutics, Paige.AI, Merck, Bayer, and Lilly Oncology and has received research funding from Loxo Oncology, Helsinn Therapeutics, Merus NV, Elevation Oncology, and Rain Therapeutics. O.L. has received advisory or consulting fees from Hologic and Janssen Research & Development, LLC. M.A. has received advisory or consulting fees from Axis Medical Education, Clinical Education Alliance, LLC, Merck Sharp & Dohme, PeerView Institute for Medical Education (PVI), Physicians' Education Resource, RMEI Medical Education, LLC, and Roche. The following Authors declare no competing interests: D.K., S.Y., S.N., K.N., R.F., N.R., I.R., J.C., and A.Y.
