## [Peer Review File · Nature Communications]

REVIEWER COMMENTS

Reviewer #1 (Remarks to the Author): Expert in cancer genomics and molecular profiling

This paper by Kim et al is a succinct and highly informative reporting of the performance of cytology samples and in the MSK-IMPACT program. The experience is extensive and highly meritorious. Reporting this is critical for others to assess- and incorporate QC steps as well as the protocols to ensure even higher accuracy. Congratulations on a fine paper that is important for the cancer community and perhaps for the emerging field of somatic mutations in noncancer outcomes.

This reviewer has a few questions and comments:

1. in the presentation of minimal DNA concentrations, what prompted the shift from 0.9 to 0.54 ng/uL? it would be helpful to provide an estimate of the number of DNA molecules/copies assayed? This is key for low 'VAF' due to low concentration of tumors.
2. The above also raises the question of subclones. did you look at this in any way? With the parallel surgical samples, it might be intriguing to include this analyses.
3. Please amplify/clarify what you mean by tumor burden. Is it NS/disruptive alterations plus or minus nonsynonymous. Please make this crystal clear as many don't fully understand the difference.
4. In the methods, why only was Genie used- this is a challenging resource for true frequencies. Seems TCGA and PCAWG might be more stable...
5. please clarify the analyses at line 331-332- "similar plus a Star Wars p value" are hard to understand....
6. for section of lines 335-340 or so- what was missed and why do you think the mutations were missed? here, please address this in the discussion as this is the key issue- false negatives which are hard to overcome and need to be considered in probability assessments.

minor point:

p15- the convention is to use italics for gene names and symbols when referring to genes

Reviewer #2 (Remarks to the Author): Expert in molecular cancer cytology

The authors investigated an impressive high number of diagnostic cytologic samples that were analyzed by a well established and acknowledged comprehensive genomic profiling assay. The results show that cytological samples are well suitable for genomic profiling. The authors also described several advancements in cytological sample preparation. The authors also found a way to deal with sample contamination which is a huge problem for cytology samples especially for cell blocks and direct smears. The results are not completely new but important for clinical management of cancer patients and truly bring forward the field of predictive cancer testing. The analysis of cytology specimen supernatant is really innovative and may change clinical workflows especially for lung and pancreatic cancer testing. The manuscript is well written and has meaningful tables and figures.

As a practicing pathologist and trained cytopathologist I recommend publishing such a comprehensive

study and make it available to the community.

Reviewer #3 (Remarks to the Author): Expert in molecular cancer cytology, liquid biopsies, and genomics

The manuscript entitled "Maximizing the clinical utility and performance of cytology samples for comprehensive genetic profiling – A report on the impact of process optimization through the analysis of 4,871 cytology samples profiled by MSK-IMPACT" highlighted that the routine use of cytologic samples for molecular testing constitute a robust approach that can deliver the same results as larger biopsy samples. Process optimization and the implementation of robust quality control processes, including contamination checks are pivotal to maximizing the yield and utility of these samples. A reassessment on how tissue blocks are processed and prepared would be an important aspect of cytology practice as a whole, to include specialized instrumentation for processing small samples without risk of cross-contamination. ScfDNA recovered from supernatants is an invaluable source of tumor derived DNA which circumvents the processing where most contamination is bound to happen in current practice, and while failure rates due to limited nucleic acid recovery are higher than tissue blocks, their use could rescue the majority of cases where high tumor is identified but FFPE material is insufficient for sequencing.

- The Authors should provide the expand forms for all acronyms, including gene acronyms, through the text when they first appear.
- Gene acronyms should be written in italics.

REVIEWER COMMENTS

Reviewer #1 (Remarks to the Author): Expert in cancer genomics and molecular profiling

This paper by Kim et al is a succinct and highly informative reporting of the performance of cytology samples and in the MSK-IMPACT program. The experience is extensive and highly meritorious. Reporting this is critical for others to assess- and incorporate QC steps as well as the protocols to ensure even higher accuracy. Congratulations on a fine paper that is important for the cancer community and perhaps for the emerging field of somatic mutations in noncancer outcomes.

This reviewer has a few questions and comments:

1. in the presentation of minimal DNA concentrations, what prompted the shift from 0.9 to 0.54 ng/uL? it would be helpful to provide an estimate of the number of DNA molecules/copies assayed? This is key for low "VAF" due to low concentration of tumors.

Response: On our initial validation of MSK-IMPACT for FFPE, we established 0.9 ng/uL as the minimum amount required to retain high sequencing performance across samples. However, with further experience after implementation, we noted that up to 50% of samples with inputs below this amount had suitable sequencing results, albeit with higher background noise. With the implementation of dual barcoding, which further optimized the error correction and performance of very limited material, we reassessed the minimal input requirements and determined that most samples with inputs as low as 30ng (but not below) had high sequencing success and retained the expected sensitivity.

Regarding the DNA molecular/copies estimate, we do not routinely assess this prior to NGS testing. However, based on ddPCR tests for single gene targets performed on FFPE samples as part of validation procedures, the average number is 238 copies per nanogram (range 14-489 copies/ng). Therefore, for samples at the lowest input of 30 ng we would be querying, on average, ~ 8,100 copies, which would robustly allow us to maintain our conservatively stated limit of detection of 2% VAF. The following has been added to the methods section to provide additional context:

(page 7; lines 156 – 161):

The change in the minimal input cutoffs was informed by the ongoing review of assay performance. As reference, based on existing validation data for MSK-IMPACT, the estimated number of target DNA molecules/copies per ng of DNA across FFPE samples of variable quality averages 238ng (range 14-489ng; 95% CI: 220.5 – 255.5), as established by digital droplet PCR assays. For samples at the lowest DNA input of 30ng, we would expect 95% of cases to have at least 6,615 molecules to query and allow robust performance of the assay.

2. The above also raises the question of subclones. did you look at this in any way? With the parallel surgical samples, it might be intriguing to include this analyses.

Response: The reviewer raises a very important point that we hope to address in a future dedicated study, incorporating additional clinical data (e.g. treatment history and tumor composition/histologic review amongst others) and follow up studies, which are beyond the scope of this work.

However, preliminary data supports that most discrepancies identified across surgical:cytology tumor pairs could be associated with the presence of subclones. It should be noted that while we made every effort to compare the same tumor in each surgical:cytology pair, there were variations in time collection across their treatment course since samples were collected and profiled based on clinical need. This introduces several variables that make this question difficult to tease out. In our review of the paired data, we stratified genetic alterations based on level of actionability and found that the overwhelming majority of driver alterations with OncoKb level 1 actionability were identified in both samples. Since driver events are often present homogeneously across the tumor, and are invariably retained across timepoints in the tumor evolution, the high concordance is expected. By contrasts, other events categorized with a low or no level of actionability, had low concordance, suggesting these are heterogeneously present in the tumor tissues sampled, either because they are passenger events that are regionally clonal or subclonal or represent clonal evolution in the later sample.

Not surprisingly, events categorized as resistance mutations had the lowest overlap, either due to the timing of the paired sample (one sample collected at baseline and other at the time of resistance) or due to known heterogeneity of resistance mutations across various regions of the tumor. This is described on Pages 16 and 17; lines 359-371 with additional edits and the addition of the below text:

It should be noted that mutations categorized in the No Level category (variants of unknown significance) constituted the largest proportion of all mutations (62%; 3,468/5,593) and were responsible for most of the discrepancies encountered. Across all non-OncoKB Level 1 categories, non-detected variants were equally distributed across surgical and cytologic samples likely reflecting a heterogenous nature (differential passenger events, acquisition of additional mutations in the time interval of the two samples), rather than false negativity in testing. Discordance in mutations categorized as Level R1 categories were primarily attributed to acquisition of resistance mechanisms in later samples, as well as heterogeneity of resistance alterations across tumors.

3. Please amplify/clarify what you mean by tumor burden. Is it NS/disruptive alterations plus or minus nonsynonymous. Please make this crystal clear as many don't fully understand the difference.

Response: A TMB calculation is performed for each sample using all non-synonymous mutations divided by the total genomic target region which is normalized to the exonic coverage of the NGS panel. This is clarified on page 8:

Tumor mutational burden (TMB) was calculated for each sample by dividing the total number of non-synonymous mutations, including driver mutations in oncogenes, by the total genomic target region of the respective MSK-IMPACT panel version. Calculated values were reported as mutations per megabase (Mb).

4. In the methods, why only was Genie used- this is a challenging resource for true frequencies. Seems TCGA and PCAWG might be more stable...

Response: We greatly appreciate the question as it also allows us to clarify some important points.

The decision to use the Genie Cohort was strategic, not only because it constituted a larger cohort (47,271 tumor samples) compared to PCAWG (n = 2,658) and TCGA (n = 10,967) but also because the data comes from clinical testing, whereas TCGA and PCAWG were done in a research setting as whole exome or whole genome sequencing (TCGA mostly whole exome; PCAWG all whole genome). Among the main aims of this work was to evaluate whether broad NGS panel testing of cytology samples identified clinically relevant alterations as effectively as non-cytology samples and if they could be robustly utilized in oncologic practice to guide disease classification and select targeted therapies. A key point that was perhaps not clear in our description is that our comparison was confined to the tumors MSKCC contributed to the GENIE project. As the highest contributor of this cohort, this accounts for 44,655 clinically sequenced samples (excludes cytology samples), which are highly curated for tumor type and with clinical information. This straight comparison eliminates the heterogeneity associated with results of smaller panels from other contributors while also aligns well with the same overall process.

We have provided clarification with the following text (page 8):

Results were compared to existing, highly curated, MSK-IMPACT sequencing data from corresponding solid tumors (non-cytology) which has been previously published as part of the American Association for Cancer Research (AACR) Project Genomics Evidence Neoplasia Information Exchange (GENIE) Pan-Cancer Cohort. In all, this encompassed results from 44,655 unique tumors with a direct focus on the clinical actionability of the alterations detected.

5. please clarify the analyses at line 331-332- "similar plus a Star Wars p value" are hard to understand....

Response: We thank the reviewer for prompting clarification on this. The sentence has been modified to the following text (page 15):

Overall, cytology samples delivered high performance with similar sequencing metrics compared to their surgical biopsy/excision counterparts. The average coverage, while high for both sample types and retaining a similar technical sensitivity, was slightly lower in cytology samples (584x vs 628x), reaching statistical significance ($p=0.00028$).

6. for section of lines 335-340 or so- what was missed and why do you think the mutations were missed? here, please address this in the discussion as this is the key issue- false negatives which are hard to overcome and need to be considered in probability assessments.

Response: The general concerns for false negativity in cytologic samples is an important issue to address. We feel that, provided sufficient tumor proportion, both surgical and cytologic samples perform equally for the detection of the most relevant alterations. Our findings show that the overwhelming majority (93%) of alterations categorized as OncoKB level 1 (strong drivers that are retained throughout the life of the tumor and that are immediately actionable), were detected in both samples. As seen in figure 5D, a small proportion of cytology samples (2%) further detected drivers not found in the surgical counterpart and vice versa (5%). The overall superiority of a surgical sample is essentially marginal when considering the risks that are incurred in a larger surgical procedure. As expected, in all of cases that showed discrepancies for the Level 1 alterations cases, this was related to low tumor content. For the cytology samples this could be easily addressed at the time of rapid onsite evaluation (ROSE) assessment to obtain more material.

As stated in question 2, our tumor normal pair sets were assembled from samples obtained for clinical necessity, with the majority of samples obtained at different timepoints across the patient's treatment course. This introduces variables related to clonal evolution, development of resistance and tumor heterogeneity with large number of passenger mutations that will generate discrepancies between the two samples. Indeed, this is supported by our findings of lowest concordance among variants of unknown significance, characterized as NO LEVEL and which encompassed the largest number of mutations "missed" by both cytologic and surgical samples. Similarly, resistance mutations were variably present in both sample types, partly due to the differential timing of the sampling but also related to tumor heterogeneity. We further highlight the fact that in the overall analysis of VAF, the cytologic samples demonstrated slightly higher VAF compared to the surgical samples, despite the

routine practice of macro-dissection for enrichment of surgical samples. This further suggests that issues of non-detection are more likely to be related to the heterogeneous presence of alterations in the tumor rather than false negativity.

This was initially mentioned on Page 16 but we have added additional language to this section to emphasize these points.

It should be noted that mutations categorized in the No Level category (variants of unknown significance) constituted the largest proportion of all mutations (62%; 3,468/5,593) and were responsible for most discrepancies encountered. Non-detected variants were equally distributed across surgical and cytologic samples, likely reflecting the heterogeneous nature of the alterations (differential passenger events, the acquisition of additional mutations in the time interval of the two samples) rather than false negativity in testing. Discordances in mutations categorized as level R1 were primarily attributed to acquisition of resistance mechanisms in the later samples or to the heterogeneous presence of resistance alterations across tumors. Further details are provided in Fig. 5d and 5h and Supplementary Table 4.

minor point:

p15- the convention is to use italics for gene names and symbols when referring to genes

Response: We thank the reviewer for pointing this out, gene names and symbols have been italicized.

Reviewer #2 (Remarks to the Author): Expert in molecular cancer cytology

The authors investigated an impressive high number of diagnostic cytologic samples that were analyzed by a well established and acknowledged comprehensive genomic profiling assay. The results show that cytological samples are well suitable for genomic profiling. The authors also described several advancements in cytological sample preparation. The authors also found a way to deal with sample contamination which is a huge problem for cytology samples especially for cell blocks and direct smears. The results are not completely new but important for clinical management of cancer patients and truly bring forward the field of predictive cancer testing. The analysis of cytology specimen supernatant is really innovative and may change clinical workflows especially for lung and pancreatic cancer testing. The manuscript is well written and has meaningful tables and figures.

As a practicing pathologist and trained cytopathologist I recommend publishing such a comprehensive study and make it available to the community.

Response: We greatly appreciate the reviewer for their consideration and comments regarding this study.

Reviewer #3 (Remarks to the Author): Expert in molecular cancer cytology, liquid biopsies, and genomics

The manuscript entitled "Maximizing the clinical utility and performance of cytology samples for comprehensive genetic profiling – A report on the impact of process optimization through the analysis of 4,871 cytology samples profiled by MSK-IMPACT" highlighted that the routine use of cytologic samples for molecular testing constitute a robust approach that can deliver the same results as larger biopsy samples. Process optimization and the implementation of robust quality control processes, including contamination checks are pivotal to maximizing the yield and utility of these samples. A reassessment on how tissue blocks are processed and prepared would be an important aspect of cytology practice as a whole, to include specialized instrumentation for processing small samples without risk of cross-contamination. ScfDNA recovered from supernatants is an invaluable source of tumor derived DNA⁷ which circumvents the processing where most contamination is bound to happen in current practice, and while failure rates due to limited nucleic acid recovery are higher than tissue blocks, their use could rescue the majority of cases where high tumor is identified but FFPE material is insufficient for sequencing.

- The Authors should provide the expand forms for all acronyms, including gene acronyms, through the text when they first appear.

- Gene acronyms should be written in italics.

Response: We thank the reviewer for pointing this out, gene names and symbols have been italicized. Acronyms have been expanded at the first mention in the text.

REVIEWERS' COMMENTS

Reviewer #1 (Remarks to the Author):

the authors have adeptly and clearly answered this reviewer's queries. This is an excellent paper that should be published.

Reviewer #2 (Remarks to the Author):

The reviewers' points are sufficiently answered.

Reviewer #3 (Remarks to the Author):

The Authors have addressed all my concerns and I have no further comments